# α8β1 integrin regulates nutrient absorption through an Mfge8-PTEN dependent mechanism

Amin Khalifeh-Soltani[1,2], Arnold Ha[1], Michael J Podolsky[1,2], Donald A McCarthy[1], William McKleroy[1], Saeedeh Azary[1], Stephen Sakuma[1], Kevin M Tharp[3,4], Nanyan Wu[5], Yasuyuki Yokosaki[6], Daniel Hart[1], Andreas Stahl[3,4], Kamran Atabai[1,2,5]*

[1]Cardiovascular Research Institute, University of California, San Francisco, San Francisco, United States; [2]Department of Medicine, University of California, San Francisco, San Francisco, United States; [3]Metabolic Biology, University of California, Berkeley, Berkeley, United States; [4]Department of Nutritional Sciences and Toxicology, University of California, Berkeley, Berkeley, United States; [5]Lung Biology Center, University of California, San Francisco, San Francisco, United States; [6]Cell-Matrix Frontier Laboratory, Biomedical Research Unit, Health Administration Center, Hiroshima University, Hiroshima, Japan

**Abstract** Coordinated gastrointestinal smooth muscle contraction is critical for proper nutrient absorption and is altered in a number of medical disorders. In this work, we demonstrate a critical role for the RGD-binding integrin α8β1 in promoting nutrient absorption through regulation of gastrointestinal motility. Smooth muscle-specific deletion and antibody blockade of α8 in mice result in enhanced gastric antral smooth muscle contraction, more rapid gastric emptying, and more rapid transit of food through the small intestine leading to malabsorption of dietary fats and carbohydrates as well as protection from weight gain in a diet-induced model of obesity. Mechanistically, ligation of α8β1 by the milk protein Mfge8 reduces antral smooth muscle contractile force by preventing RhoA activation through a PTEN-dependent mechanism. Collectively, our results identify a role for α8β1 in regulating gastrointestinal motility and identify α8 as a potential target for disorders characterized by hypo- or hyper-motility.

*For correspondence: kamran.atabai@ucsf.edu

**Competing interests:** The authors declare that no competing interests exist.

## Introduction

Coordinated gastrointestinal motility orchestrates nutrient absorption by mixing ingested food with digestive enzymes (*Armand et al., 1994*; *Kong and Singh, 2008*) and by propelling the food bolus from the stomach to the small intestine, the primary site of nutrient absorption. Dysfunctional gastrointestinal motility occurs in a number of common disease processes (*Ambartsumyan and Rodriguez, 2014*; *Fan and Sellin, 2009*), is difficult to treat, and is characterized by either accelerated motility leading to rapid intestinal transit times, diarrhea, and malabsorption (*Spiller, 2006*) or delayed motility leading to nausea, vomiting, and aspiration of stomach contents (*Enweluzo and Aziz, 2013*; *Janssen et al., 2013*). A better understanding of the molecular mechanisms that regulate gastrointestinal motility has significant clinical implications for disorders characterized by hypo- or hyper-motility.

Milk fat Globule Epidermal Growth Factor like 8 (Mfge8) is an integrin ligand (*Atabai et al., 2009*) that is highly expressed in breast milk (*Atabai et al., 2005*). Mfge8 coordinates absorption of

**eLife digest** Animals absorb nutrients from the food they eat in a complicated process that involves multiple steps. In the mouth, teeth break down the food into smaller chunks. Then the food passes through the stomach and small intestine, where enzymes break it down into individual molecules that are small enough to be absorbed by cells that line the small intestine. These cells then package the molecules and release them into the bloodstream so that they can be distributed to the rest of the body.

Muscles in the wall of the small intestine control how quickly food travels through this part of the gut. If food moves too quickly, the cells that line the intestine have less time to absorb the food molecules and may fail to absorb enough nutrients. If the food moves too slowly, an individual may experience nausea or vomiting, or the contents of their stomach may spill into their lungs.

In 2014, researchers reported that a protein in breast milk called Mfge8 helps to boost the number of fat molecules absorbed from food. Now, Khalifeh-Soltani et al. – including some of the same researchers involved in the earlier work – show that Mfge8 also slows the rate at which food travels through the small intestine in mice. Mfge8 binds to another protein called integrin α8β1 to control how often the intestine muscles contract. Genetically engineered mice that lacked integrin α8β1 developed diarrhea and food passed through their intestines more quickly than in normal mice. Furthermore, these mice did not gain as much weight as normal mice when they were fed a high-fat diet.

Khalifeh-Soltani et al.'s findings show that Mfge8 has a dual role in controlling the absorption of food molecules in the small intestine. The next challenge is to find out whether drugs that alter the activity of integrin α8β1 could be used to help treat patients with diseases in which food moves too quickly, or too slowly, through the gut.

dietary fats by promoting enterocyte fatty acid uptake after ligation of the αvβ3 and αvβ5 integrins (*Khalifeh-Soltani et al., 2014*). Mfge8 also modulates smooth muscle contractile force (*Kudo et al., 2013*). In mice deficient in Mfge8 (*Mfge8$^{-/-}$*), airway and jejunal smooth muscle contraction are enhanced in response to contractile agonists after these muscle beds have been exposed to inflammatory cytokines but not under basal conditions (*Kudo et al., 2013*). Contraction of antral smooth muscle is a key determinant of the rate at which a solid food bolus exits the stomach and transits through the primary site of nutrient absorption, the small intestine (*Haba and Sarna, 1993*; *Kelly, 1980*; *Burks et al., 1985*). Since Mfge8 promotes enterocyte fatty acid uptake and can regulate smooth muscle contraction, we were interested in examining whether Mfge8 reduces the force of basal antral smooth muscle contraction, thereby slowing gastrointestinal motility and allowing a greater time for nutrient absorption.

α8β1 is a member of the RGD-binding integrin family that is prominently expressed in smooth muscle (*Zargham et al., 2007*; *Zargham and Thilbault, 2006*; *Schnapp et al., 1995*). The most definitive in vivo role described for α8β1 is in kidney morphogenesis where deletion of this integrin subunit leads to impaired recruitment of mesenchymal cells into epithelial structures (*Müller et al., 1997*; *Humbert et al., 2014*). Osteopontin, fibronectin, vitronectin, nephronectin, and tenascin-C have all previously been identified as ligands for α8β1 (*Schnapp et al., 1995*; *Denda et al., 1998*; *Kiyozumi et al., 2012*). In this work we show that Mfge8 is a novel ligand for α8β1 and that Mfge8 ligation of α8β1 reduces the force of gastric antral smooth muscle contraction, the extent of gastric emptying, and the rate at which a food bolus transits through the small intestine. We further show that mice with smooth muscle-specific deletion of α8 integrin subunit (*Itga8$^{flox/flox}$—Tg(Acta2-rtTA, TetO-Cre)*) fail to properly absorb ingested fats and carbohydrates and are partially protected from weight gain in a model of diet-induced obesity. α8β1 slows gastrointestinal motility by increasing the activity of Phosphatase and tensin homolog (PTEN) leading to reduced activation of the Ras homolog gene family member A (RhoA).

# Results

## Mfge8 regulates gastrointestinal motility

To determine whether Mfge8 regulates the force of antral smooth muscle contraction, we isolated gastric antral rings and measured the force of antral contraction in a muscle bath. Antral rings isolated from $Mfge8^{-/-}$ mice had increased force of contraction in response to both methacholine (MCh) and KCl as compared with wild type (WT) controls (*Figure 1A and B*). The thickness of antral smooth muscle was not different when comparing Mfge8$^{-/-}$ and WT mice indicating that the enhanced contraction was not due to smooth muscle hypertrophy (*Figure 1—figure supplement 1A and B*). Incubation with recombinant Mfge8 (rMfge8), but not a recombinant construct where the integrin-binding RGD sequence was mutated to RGE, rescued enhanced contraction indicating that the effect of Mfge8 on gastric smooth muscle was integrin-dependent (*Figure 1A,B*). Induction of Mfge8 expression in the smooth muscle of $Mfge8^{-/-}$—$Tg(Acta2-rtTA, TetO-Mfge8)$ transgenic mice, abbreviated Mfge8$^{-/-}$sm$^{+}$, where Mfge8 expression was driven by a tetracycline-inducible Mfge8 transgene coupled with an α-smooth muscle-rtTA transgene (*Figure 1—figure supplement 1C*) also rescued enhanced contraction (*Figure 1C*; *Figure 1—figure supplement 1D*). Of note, unlike antral rings, duodenal rings from $Mfge8^{-/-}$ mice did not have enhanced contraction (*Figure 1—figure supplement 1E*) consistent with our previously reported findings in jejunal smooth muscle rings

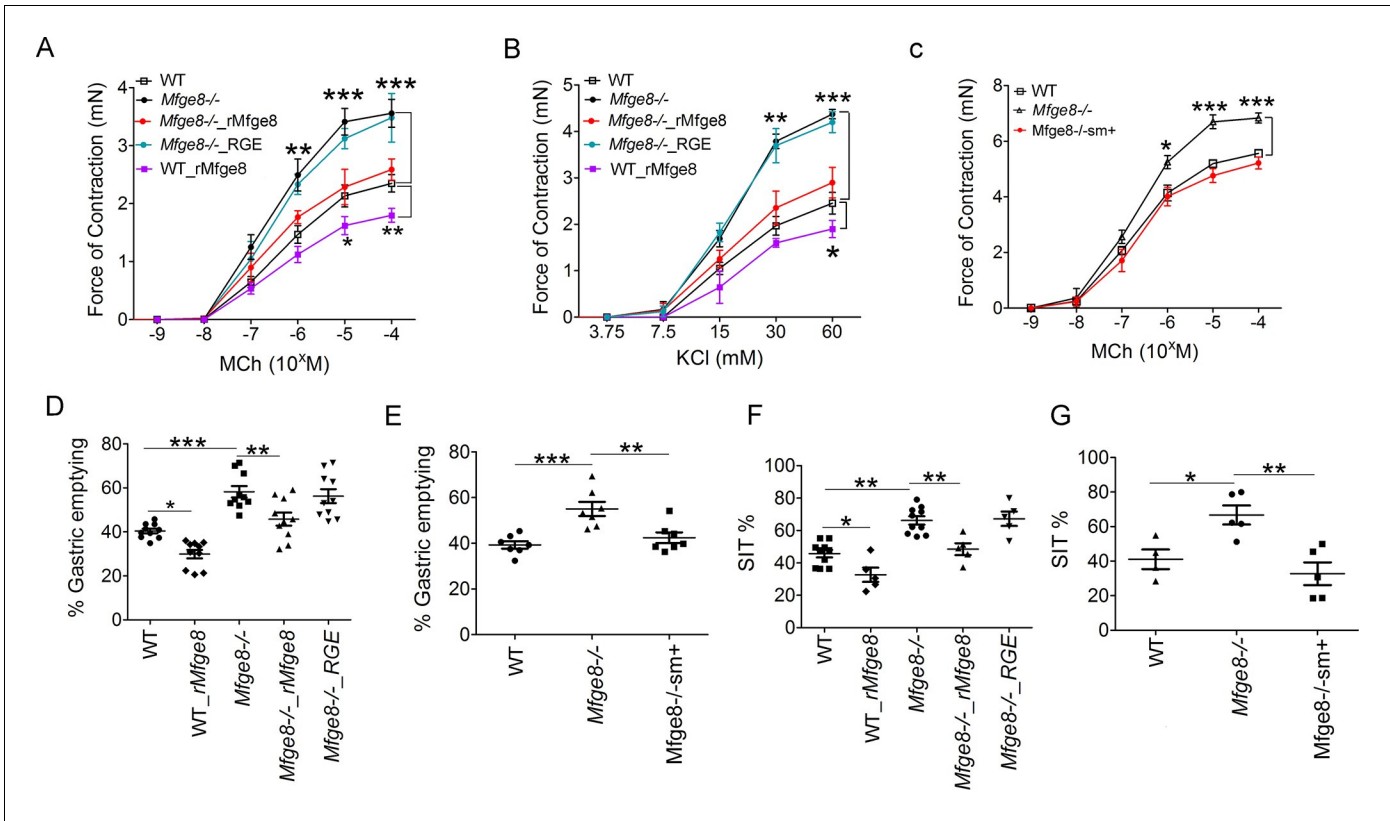

**Figure 1.** Mfge8 regulates gastrointestinal motility. (A–C) Force of antral smooth muscle ring contraction with and without the addition of rMfge8 or RGE construct in $Mfge8^{-/-}$ and WT in response to MCh (A, N = 4–5) or KCl (B, N = 4–5) or after in vivo induction of smooth muscle Mfge8 expression in Mfge8$^{-/-}$sm+ mice in response to MCh (C, N = 5). (D, E) The rate of gastric emptying in $Mfge8^{-/-}$ and WT with and without the addition of rMfge8 or RGE construct (D, N = 10) or after smooth muscle transgenic (Mfge8-/-sm+) expression of Mfge8 (E, N = 7). (F–G) Small intestinal transit time in $Mfge8^{-/-}$ and WT with and without the addition of rMfge8 or RGE construct (F, N = 5–10) or after smooth muscle transgenic expression of Mfge8 (G, N = 4–5). Female mice were used for all experiments. *p<0.05, **p<0.01, ***p<0.001. Data are expressed as mean ± s.e.m.

The following figure supplement is available for figure 1:

**Figure supplement 1.** Mfge8$^{-/-}$ smooth muscle morphology and smooth muscle expression of Mfge8 in Mfge8$^{-/-}$sm+ mice.

(*Kudo et al., 2013*). We next determined whether enhanced antrum contractility was associated with altered gastric emptying and small intestinal transit times (SIT), two in vivo measures of gastro-intestinal motility. *Mfge8*[-/-] mice had significantly more rapid gastric emptying and more rapid SIT (*Figure 1D–G*). Administration of rMfge8 by gavage and transgenic smooth muscle expression of Mfge8 significantly reduced the rate of gastric emptying and prolonged SIT in *Mfge8*[-/-] mice (*Figure 1D–G*). Administration of rMfge8 by gavage also significantly reduced gastric emptying and prolonged SIT in WT mice (*Figure 1D and F*).

## Mfge8 is a ligand for the α8β1 integrin

The αvβ3 and αvβ5 integrins are the known cell surface receptors for Mfge8 (*Atabai et al., 2005*; *Hanayama et al., 2004*; *Hanayama et al., 2002*) and mediate the effect of Mfge8 on fatty acid uptake (*Khalifeh-Soltani et al., 2014*). We therefore investigated whether these integrins mediated the effect of Mfge8 on gastrointestinal motility. Antrum contraction was similar in WT, *Itgb3*[-/-], *Itgb5*[-/-] and *Itgb3*[-/-]::*Itgb5*[-/-] mice (*Figure 2—figure supplement 1A*). Gastric emptying and SIT were also similar in *Itgb3*[-/-], *Itgb5*[-/-] and *Itgb3*[-/-]::*Itgb*[-/-] mice and rMfge8 significantly reduced the rate of gastric emptying and prolonged SIT in each mouse line (*Figure 2—figure supplement 1B and C*). These data indicate that the effect of Mfge8 on smooth muscle contraction occurs via a novel RGD-binding, integrin partner.

We have previously shown that Mfge8 does not bind the RGD-binding integrins αvβ6, αvβ8, and α5β1 (*Atabai et al., 2009*), leaving α8β1 and αvβ1 as the potential RGD-binding receptors for the effect of Mfge8 on smooth muscle contraction. We initially focused on α8β1 because of its high expression in smooth muscle (*Schnapp et al., 1995*; *Kitchen et al., 2013*). To determine if α8β1 is a receptor for Mfge8, we used a solid-phase assay to analyze the direct binding of Mfge8 to purified α8. We included purified αvβ3 and α5β1 as positive and negative controls, respectively. Mfge8 bound to α8β1 and αvβ3, but not to α5β1 (*Figure 2A*). To further confirm this interaction, we evaluated cell adhesion of SW480 cells, a human colon cancer cell line, transfected with α8 or β3 to Mfge8 (*Figure 2B*). Control SW480 cells express the Mfge8 ligand αvβ5 as well as α5β1 and bind Mfge8 in an αvβ5-dependent manner. Mock-transfected control SW480 cells adhered to Mfge8 and adherence was blocked by anti-β5 antibody (ALULA). In the presence of ALULA, β3-transfected cells adhered to Mfge8 (αvβ3 is a known receptor for Mfge8, as above), and adherence was blocked by an anti-β3 antibody (LM609). α8-transfected SW480 cells also adhered to Mfge8 in the presence of ALULA, and adherence was blocked by the addition of an α8 blocking antibody (YZ3) (*Nishimichi et al., 2015*). The YZ3 antibody blocks both human and mouse α8. These results indicate that α8β1 specifically mediates cell adhesion to Mfge8. Next we analyzed adhesion of α8-transfected SW480 cells to Mfge8 at various concentrations in the presence of ALULA (*Figure 2C*). The α8-trans-fected cells adhered to Mfge8 in a dose-dependent fashion. Of note, expression of β5, β3, and α8, as evaluated by flow cytometry (*Figure 2—figure supplement 2*) and the extent of binding to Mfge8 (*Figure 2B and C*) were similar in these cell lines.

To confirm that these findings were relevant in smooth muscle cells, we evaluated adhesion of primary human gastric smooth muscle cells to Mfge8. Primary human gastric smooth muscle cells expressed the β5, β1, αv, and α8 integrin subunits (*Figure 2D*) and adhered to Mfge8 (*Figure 2E*). Adherence was significantly inhibited by blocking antibodies to the β5, β1, αv, and α8 subunits but not the α5 integrin (*Figure 2E*). Simultaneous blockade of the αv and α8 integrins had a significantly greater effect on adhesion than blocking each integrin individually (*Figure 2E*). To determine whether Mfge8 and the α8integrin colocalize in gastric smooth muscle, we injected *Mfge8*-/- mice with our recombinant Mfge8 protein containing a human FC domain, prepared tissue sections, and probed with an anti-Fc and anti-α8 antibody. As shown in *Figure 2F*, Mfge8 and α8 colocalized in gastric smooth muscle (*Figure 2F*).

## α8β1 integrin mediates the effect of Mfge8 on motility

To evaluate whether α8β1 mediates the effect of Mfge8 on gastric smooth muscle, we created a *Itga8*[flox/flox]—*Tg(Acta2-rtTA, TetO-Cre)* transgenic mouse line, *abbreviated as* α8sm[-/-], containing α8 floxed/floxed alleles, a tetracycline-inducible Cre transgene and an α-smooth muscle-rtTA transgene. The addition of doxycycline chow resulted in smooth muscle-specific recombination of α8 (*Figure 2—figure supplement 3A–C*). Gastric antral smooth muscle from α8sm[-/-] mice had enhanced

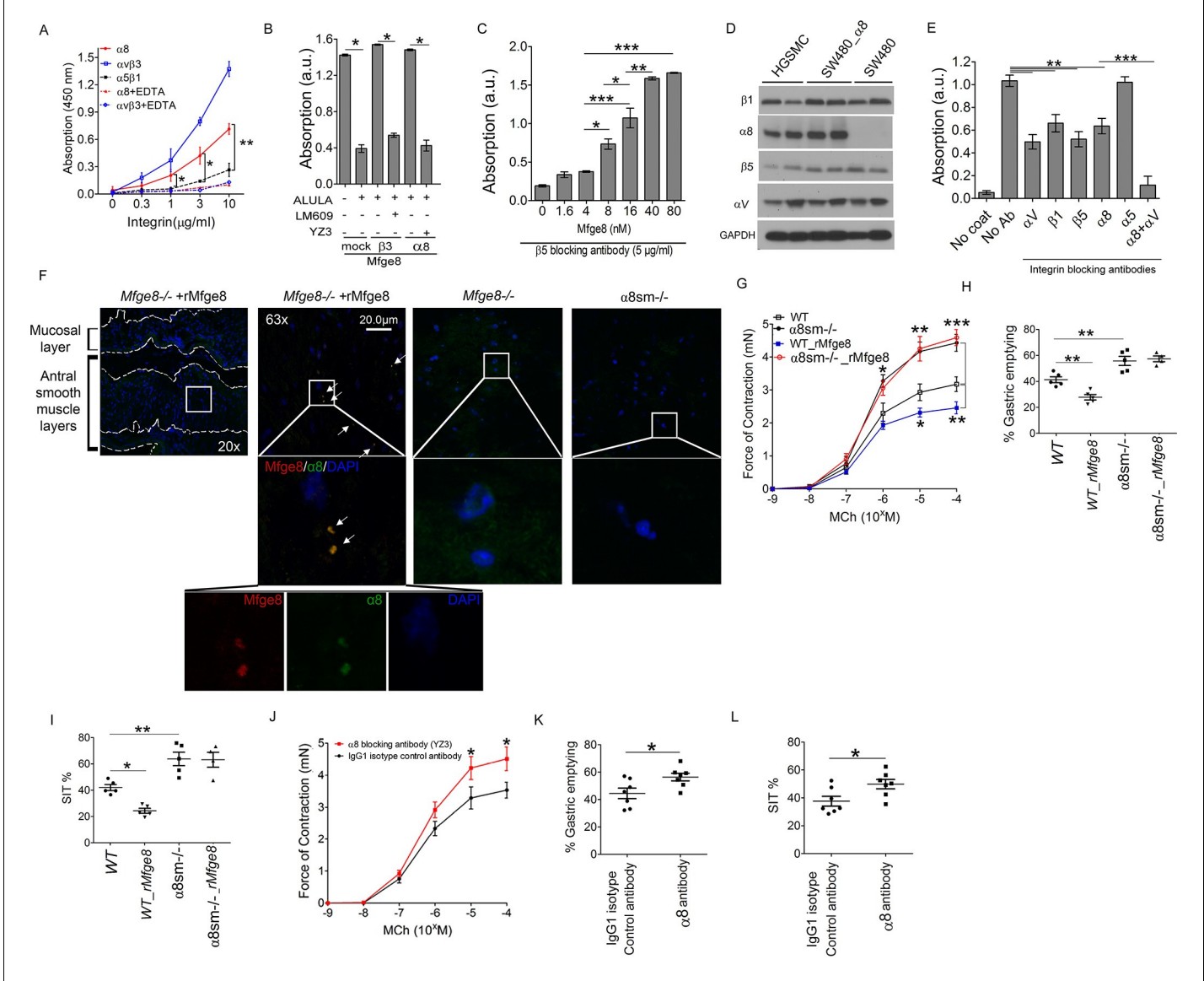

**Figure 2.** Mfge8 binds to α8 integrin to regulate gastrointestinal motility. (A) Purified α8, αvβ3, or α5β1 were used for solid-phase binding assays with purified Mfge8 at indicated concentrations in the presence or absence of 10 mM EDTA. (B) Adhesion of SW480 (mock), α8 transfected SW480 cells (α8) or β3 transfected SW480 cells (β3) adhesion to wells coated with rMfge8 (5 µg/ml) in the presence or absence of integrin blocking antibodies (5 µg/ml) against β5 (ALULA), β3 (LM609) or α8 (YZ83). (C) Dose-dependent binding of SW480 cells to wells coated with a dose range of rMfge8 in the presence of a β5 blocking antibody. (D) Western blot of integrin expression in human gastric smooth muscle cells (HGSMC), SW480 cells and α8 transfected SW480 (SW480_α8) cells. (E) Human gastric smooth muscle cell adhesion to rMfge8-coated wells in the presence of blocking antibodies against the αv, β1, β5, α8, or α5 integrin subunits. (F) Immunofluorescence staining of antral sections from *Mfge8*[-/-] and α8sm[-/-] mice with or without rMfge8 gavage proved for α8 (green), human-FC-Mfge8 recombinant construct (red) and DAPI (blue). Arrows point co-localization of Mfge8 and α8. (G) Force of antral contraction in WT and α8sm[-/-] mice in response to MCh (N = 3–4). (H) The rate of gastric emptying in α8sm[-/-] and WT mice with and without the addition of rMfge8 (N = 4–5). (I) Small intestinal transit time in α8sm[-/-] and WT mice with and without the addition of rMfge8 (N = 4–5). (J) Force of antral contraction in WT mice after IP injection of α8 blocking or control antibody in response to MCh (N = 4–5). (K) The rate of gastric emptying in WT mice after IP injection of α8 blocking or IgG1 isotype control antibody (N = 7). (L) Small intestinal transit time in WT mice after IP injection of α8 blocking or IgG1 isotype control antibody (N = 7). Female mice were used in G, H and I and male mice were used for all remaining panels. *p<0.05, **p<0.01, ***p<0.001. Data are expressed as mean ± s.e.m.

The following figure supplements are available for figure 2:

**Figure supplement 1.** Normal gastrointestinal motility in *Itgb3*[-/-], *Itgb5*[-/-] and *Itgb3*[-/-]::*Itgb5*[-/-] mice.

*Figure 2 continued*

**Figure supplement 2.** Integrin expression levels in SW480 cells.

**Figure supplement 3.** Enhanced antral contraction in α8sm[-/-] mice.

contraction in response to MCh and KCl (*Figure 2G* and *Figure 2—figure supplement 3D*). Unlike WT samples, rMfge8 did not significantly reduce the force of contraction in α8sm[-/-] antral smooth muscle (*Figure 2G* and *Figure 2—figure supplement 3D*). α8sm[-/-] mice had enhanced gastric emptying and more rapid SIT (*Figure 2H and I*). Oral gavage with rMfge8 did not significantly slow gastric emptying or prolong SIT in α8sm[-/-] mice (*Figure 2H and I*). Administration of α8 blocking antibody to WT mice significantly increased the force of antral contraction, accelerated gastric emptying, and reduced SIT (*Figure 2J–L*). In sum, these data indicate that disruption of α8β1 integrin signaling accelerates gastrointestinal motility.

## α8β1 integrin inhibits smooth muscle contractility by reducing calcium sensitivity

Enhanced antral smooth muscle contraction could be the result of an increase in the frequency of intracellular calcium oscillations after release of calcium from intracellular sources or the result of an increase in calcium sensitivity due to inactivation of the enzyme myosin light chain phosphatase (*Kudo et al., 2013*; *Bhattacharya et al., 2014*; *Kudo et al., 2012*; *Somlyo and Somlyo, 2000*; *Somlyo and Somlyo, 2003*). Antral rings from *Mfge8[-/-]* and α8sm[-/-] mice had exaggerated contraction to both MCh and KCl suggesting altered calcium sensitivity as the mechanism by which Mfge8 reduced contraction since these agonists increase intracellular calcium through different mechanisms. KCl works primarily by inducing opening of voltage-gated calcium channels leading to influx of extracellular calcium while MCh induces release of intracellular calcium stores after receptor binding. To determine whether enhanced antral contraction was due to an increase in smooth muscle calcium sensitivity, we assessed the phosphorylation status of the regulatory subunit of myosin light chain phosphatase, MYPT, and myosin light chain (MLC) (*Kudo et al., 2013*; *Bhattacharya et al., 2014*; *Kudo et al., 2012*; *Somlyo and Somlyo, 2000*; *Somlyo and Somlyo, 2003*). Antral smooth muscle from *Mfge8[-/-]* and α8sm[-/-] mice had increased phosphorylation of both MYPT and MLC in response to MCh as compared with WT smooth muscle (*Figure 3A and B*). Antral smooth muscle from *Itgb3[-/-]::Itgb5[-/-]* mice did not have increased phosphorylation of MYPT or MLC as compared with WT samples and MYPT and MLC phosphorylation that was present in response to MCh was reduced to a similar extent by rMfge8 in WT and *Itgb3[-/-]::Itgb5[-/-]* mice (*Figure 3—figure supplement 1*).

The small GTPase RhoA is a prominent regulator of MYPT phosphorylation and inhibition of RhoA has been shown to reduce the force of gastric smooth muscle contraction (*Ratz et al., 2002*; *Tomomasa et al., 2000*; *Büyükafşar and Levent, 2003*). RhoA activation, assessed by a GST pull-down assay, was significantly increased in *Mfge8[-/-]* and α8sm[-/-] antral smooth muscle as compared with WT controls while total RhoA protein expression was unchanged (*Figure 3C and D*). rMfge8 reduced RhoA activation in WT and *Mfge8[-/-]* antrum but not α8sm[-/-] antral smooth muscle (*Figure 3C and D*). Pharmacological inhibition of ROCK, the kinase downstream of RhoA responsible for phosphorylation and inactivation of MYPT, with Y-27632, inhibited antral contraction in both WT and *Mfge8[-/-]* smooth muscle reducing *Mfge8[-/-]* antral contraction to WT levels (*Figure 3E*). Intraperitoneal (IP) administration of Y-27632 also reduced gastric emptying and prolonged SIT in WT and *Mfge8[-/-]* mice with a relatively greater effect in *Mfge8[-/-]* mice (*Figure 3F and G*). Taken together, these data indicate that in gastric antral smooth muscle, the absence of α8β1 integrin results in enhanced RhoA activation leading to increased smooth muscle calcium sensitivity, antral contraction, gastric emptying, and more rapid small intestinal transit times.

## α8β1 integrin opposes PI3 kinase activity

PI3 kinase (PI3K) is a positive regulator of smooth muscle contraction (*Wang et al., 2006*; *Kawabata et al., 2000*). To determine whether the Mfge8-α8β1 axis modulates smooth muscle contraction through PI3K, we incubated antral smooth muscle rings with the PI3K inhibitor wortmannin.

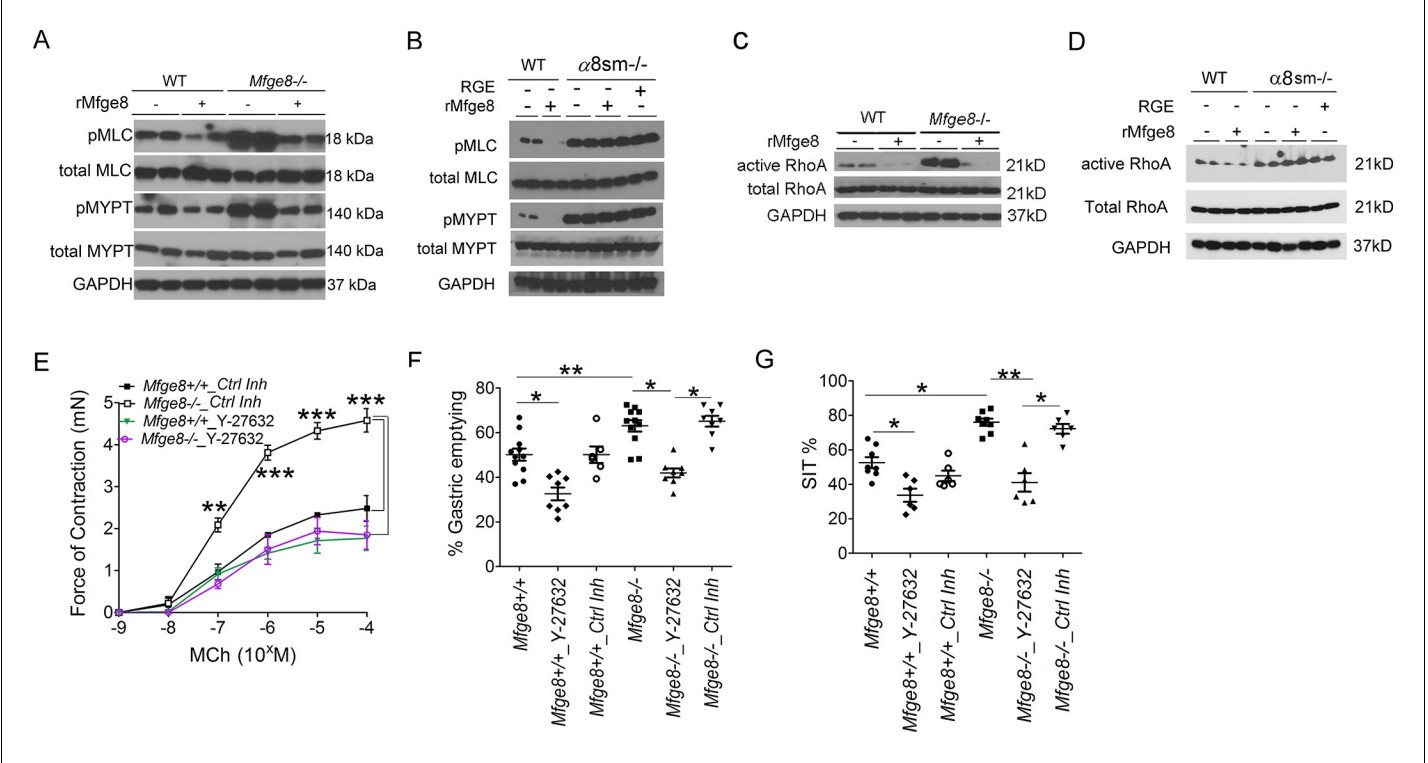

**Figure 3.** α8 integrin regulates antrum smooth muscle calcium sensitivity by preventing RhoA activation. (**A, B**) Western blot of antrum muscle strips obtained from (**A**) *Mfge8⁻/⁻* and (**B**) α8sm⁻/⁻ mice and incubated with MCh. (**C,D**) Western blot of antrum smooth muscle strips obtained from (**C**) *Mfge8⁻/⁻* and (**D**) α8sm⁻/⁻ treated with MCh demonstrating active and total RhoA using a GST pull-down assay. (**E**) Force of antral smooth muscle ring contraction with and without the addition of ROCK inhibitor Y-27632 (N = 3–4). (**F**) The rate of gastric emptying in *Mfge8⁻/⁻* and WT with and without the IP injection of ROCK inhibitor (Y-27632) or control inhibitor (N = 5–11). (**G**) Small intestinal transit times *Mfge8⁻/⁻* and WT with and without IP injection of ROCK inhibitor (Y-27632) or control inhibitor (N = 6–11). Female mice were used for all experiments. *$p<0.05$, **$p<0.01$, ***$p<0.001$. Data are expressed as mean ± s.e.m.

The following figure supplement is available for figure 3:

**Figure supplement 1.** Normal calcium sensitivity in *Itgb3⁻/⁻::Itgb5⁻/⁻* antrum smooth muscle.

Wortmannin significantly reduced contraction in *Mfge8⁻/⁻*, α8sm⁻/⁻, and WT antral smooth with a proportionally greater effect on antrum from *Mfge8⁻/⁻* and α8sm⁻/⁻ as compared with antrum from WT mice (*Figure 4A and B*). PI3K activation leads to phosphorylation of AKT. Antral rings from *Mfge8⁻/⁻* and α8sm⁻/⁻ mice had enhanced phosphorylation of AKT at serine 473 (*Figure 4C and D*). rMfge8 reduced AKT phosphorylation in *Mfge8⁻/⁻* but not α8sm⁻/⁻ samples (*Figure 4C and D*). Wortmannin also prevented the enhanced RhoA activation in *Mfge8⁻/⁻* and α8sm⁻/⁻ antral smooth muscle (*Figure 4E*).

Phosphatase and tensin homolog (PTEN) is the major negative regulator of PI3K (*Leevers et al., 1999*). To determine whether Mfge8 ligation of α8β1 opposed PI3K activation through modulation of PTEN, we measured PTEN activity using an ELISA that quantifies the ability of lysates to convert PI (3,4,5) $P_3$ to PI (4,5) $P_2$ (*Maehama and Dixon, 1998*). PTEN activity was reduced in both *Mfge8⁻/⁻* and α8sm⁻/⁻ antral rings (*Figure 5A and B*). rMfge8 significantly increased PTEN activity in antrum from WT and *Mfge8⁻/⁻* mice with no effect on antrum from α8sm⁻/⁻ mice (*Figure 5A and B*). In both WT and *Mfge8⁻/⁻* mice there was a significant inverse correlation between the extent of PTEN activity and the rate of gastric emptying and SIT (*Figure 5C and D*). rMfge8 also increased PTEN activity in primary human gastric smooth muscle cells, an effect that was blocked using a blocking antibody to α8 but not to α5 or β5 integrin subunits (*Figure 5E*). Of note, treatment with fibronectin or vitronectin, both ligands of α8β1, did not increase PTEN activity suggesting a specific effect for Mfge8 (*Figure 5—figure supplement 1*). IP administration of α8 blocking antibody also decreased antral PTEN

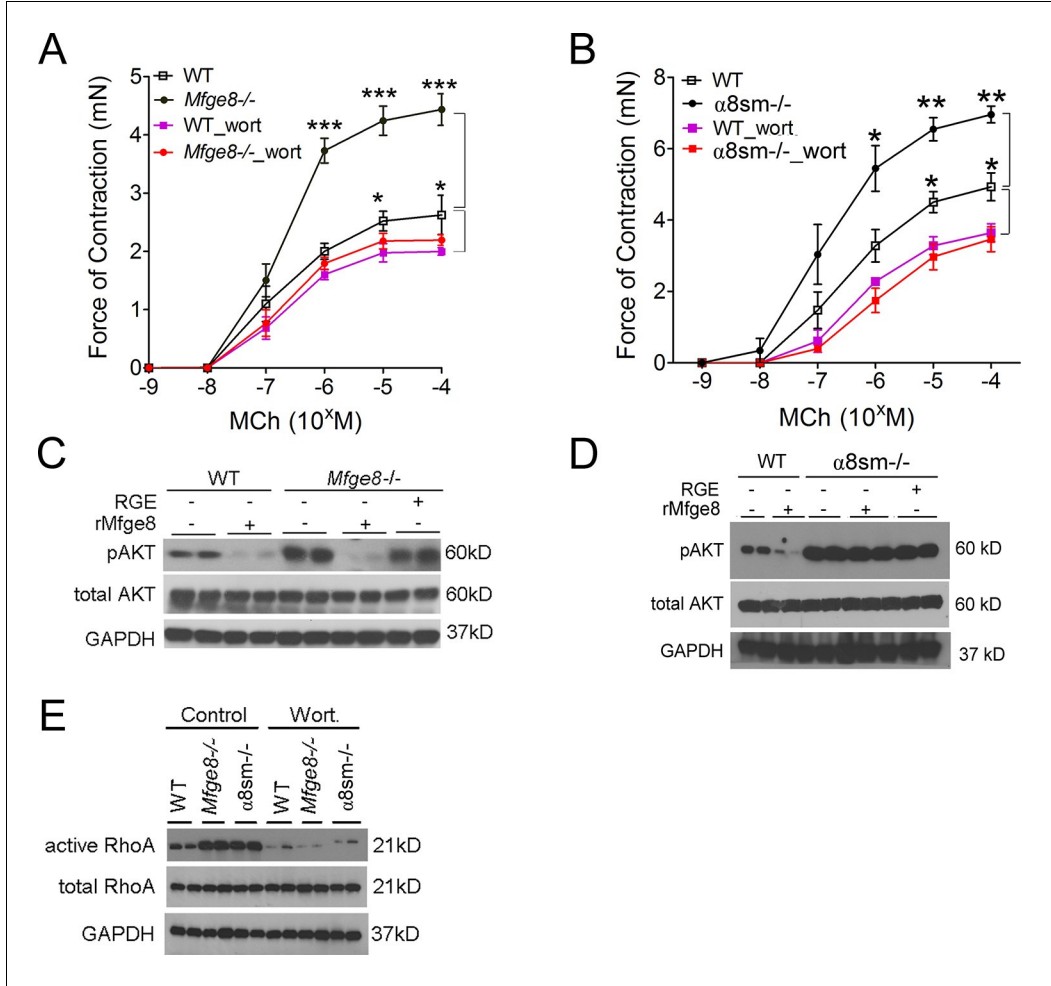

**Figure 4.** Mfge8 ligation of α8β1 integrin inhibits PI3 kinase activity. (**A–B**) Force of antral smooth muscle ring contraction with and without the addition of PI3K inhibitor wortmannin (wort 100 ng/ml) in response to MCh in WT and $Mfge8^{-/-}$ (A, N = 4–5) or WT and α8sm$^{-/-}$ (B, N = 4–5). (**C–D**) Western blot of antrum muscle strips obtained from WT and $Mfge8^{-/-}$ (**C**) and WT and α8sm$^{-/-}$ mice (**D**) incubated with MCh. (**E**) Western blot of antrum from WT, $Mfge8^{-/-}$ and α8sm$^{-/-}$ treated with wortmannin (100 ng/ml) and MCh demonstrating active and total RhoA using a GST pull-down assay. Male mice were used in panel A and B. The remaining panels include both male and female mice. *p<0.05, **p<0.01, ***p<0.001. Data are expressed as mean ± s.e.m.

activity in WT mice (*Figure 5F*). We next used siRNA to knockdown PTEN expression in primary human gastric smooth muscle cells (*Figure 5—figure supplement 2*) to evaluate the effect on smooth muscle calcium sensitivity. PTEN knockdown lead to increased MLC and MYPT phosphorylation in response to 5-HT (*Figure 5G*) as well as to increased RhoA activation (*Figure 5H*). Unlike control samples, rMfge8 did not reduce the degree of MYPT or MLC phosphorylation or RhoA activation in human gastric smooth muscle after PTEN knockdown (*Figure 5G and H*). These data indicate that α8β1 prevents RhoA activation in gastric smooth muscle by increasing the activity of PTEN.

## α8 integrin promotes nutrient absorption

We next wanted to evaluate the functional consequences of altered motility on nutrient absorption in α8sm$^{-/-}$ mice. Since we have previously reported impaired fat absorption in $Mfge8^{-/-}$ mice (*Khalifeh-Soltani et al., 2014*), we first assessed the ability of α8sm$^{-/-}$ mice to absorb dietary fats. After an olive oil gavage, α8sm$^{-/-}$ mice had significantly higher fecal triglyceride (TG) concentrations (*Figure 6A*) as well as lower serum TG levels (*Figure 6B*) as compared with WT mice. Fecal TG levels

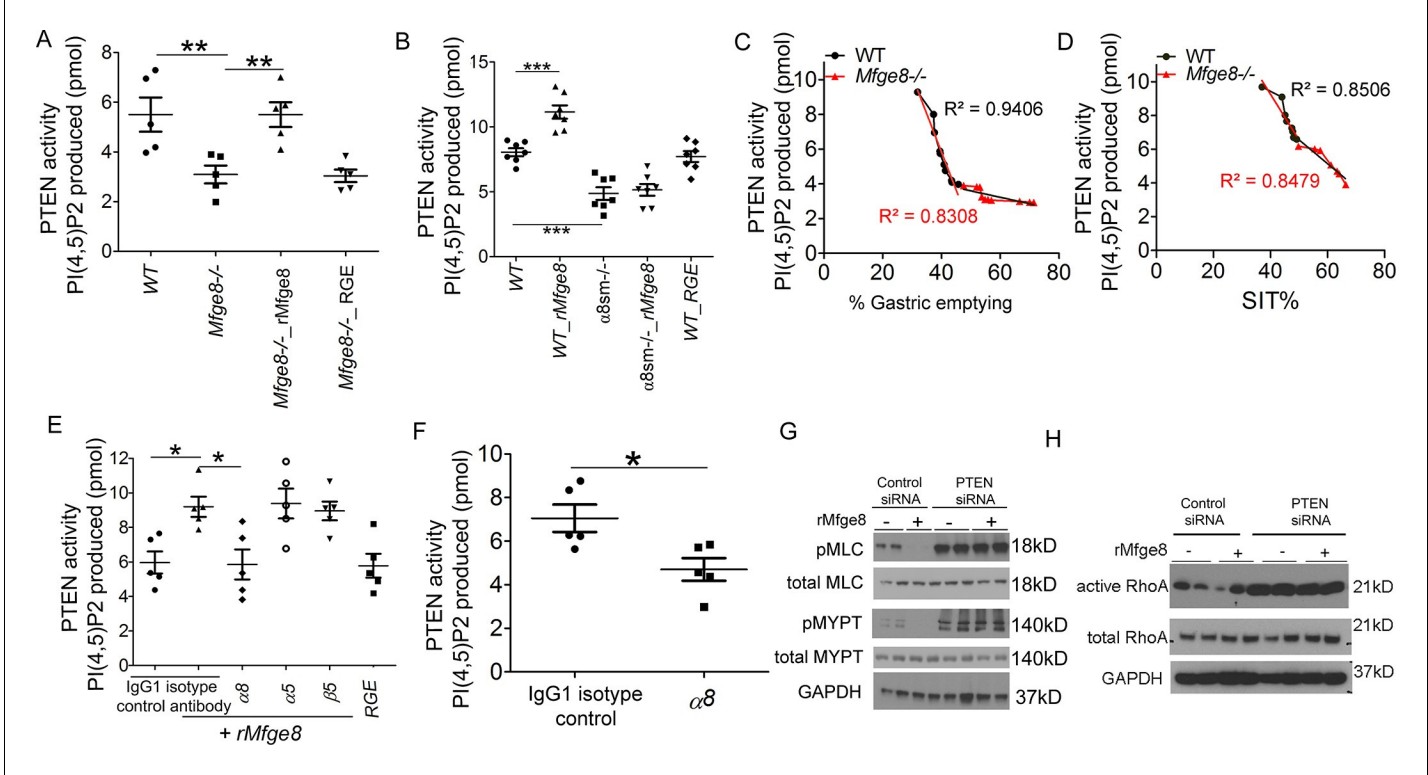

**Figure 5.** Mfge8 modulates PTEN activity. (A–B) PTEN activity in antral smooth muscle of WT and *Mfge8*[-/-] (A, N = 5) and WT and α8sm[-/-] (B, N = 7) with and without the addition of rMfge8 and RGE construct. (C–D) Correlation between the PTEN activity and the rate of gastric emptying (C, N = 11) and the small intestinal transit time in WT mice (D, N = 13). (E) PTEN activity in antral smooth muscle strips with addition of rMfge8 in presence of blocking antibody against α8, α5 and β5. (F) PTEN activity in antral smooth muscle strips of WT mice after IP injection of α8 blocking or IgG1 isotype control antibody. (N = 5). (G) Western blot in human gastric smooth muscle cells (HGSMC) treated with siRNA targeting PTEN with or without rMfge8 and then treated with 5-HT (100 μM). (H) Western blot of human gastric smooth muscle cells (HGSMC) treated with PTEN siRNA and with 5-HT demonstrating active and total RhoA using a GST pull-down assay. Both male and female mice were used for these experiments. *p<0.05, **p<0.01, ***p<0.001. Data are expressed as mean ± s.e.m.

The following figure supplements are available for figure 5:

**Figure supplement 1.** Mfge8 increases PTEN activity.

**Figure supplement 2.** siRNA knockdown of PTEN.

were also significantly higher in α8sm[-/-] mice on a normal chow diet control (CD) as compared with WT mice (*Figure 6C*). Of note, primary enterocytes isolated from α8sm[-/-] mice did not have a defect in fatty acid uptake indicating that the increase in stool fat was unlikely to be due to a defect in enterocyte fatty acid uptake (*Figure 6D*). Furthermore, IP injection of olive oil resulted in similar serum TG levels in α8sm[-/-] mice as compared with WT mice (*Figure 6E*) indicating that clearance of lipids by tissue outside of the intestinal tract was preserved in α8sm[-/-] mice. Taken together, these data indicate that α8sm[-/-] mice develop steatorrhea.

To evaluate whether malabsorption was specific for fat or represented a more generalized impairment of nutrient uptake, we measured stool glucose levels after gavage with a 2-(N-(7-Nitrobenz-2-oxa-1,3-diazol-4-yl)Amino)-2-Deoxyglucose (2NBDG) fluorescent glucose analog. 2NBDG was mixed with methylcellulose to form a semisolid bolus sensitive to antral contraction. α8sm[-/-] mice had increased stool glucose levels (*Figure 6F*) coupled with reduced enterocyte glucose levels (*Figure 6G*). Enterocytes isolated from α8sm[-/-] mice and cultured in vitro did not have a defect in 2NBDG uptake (*Figure 6H*). *Mfge8*[-/-] mice also had increased stool 2NBDG and reduced enterocyte

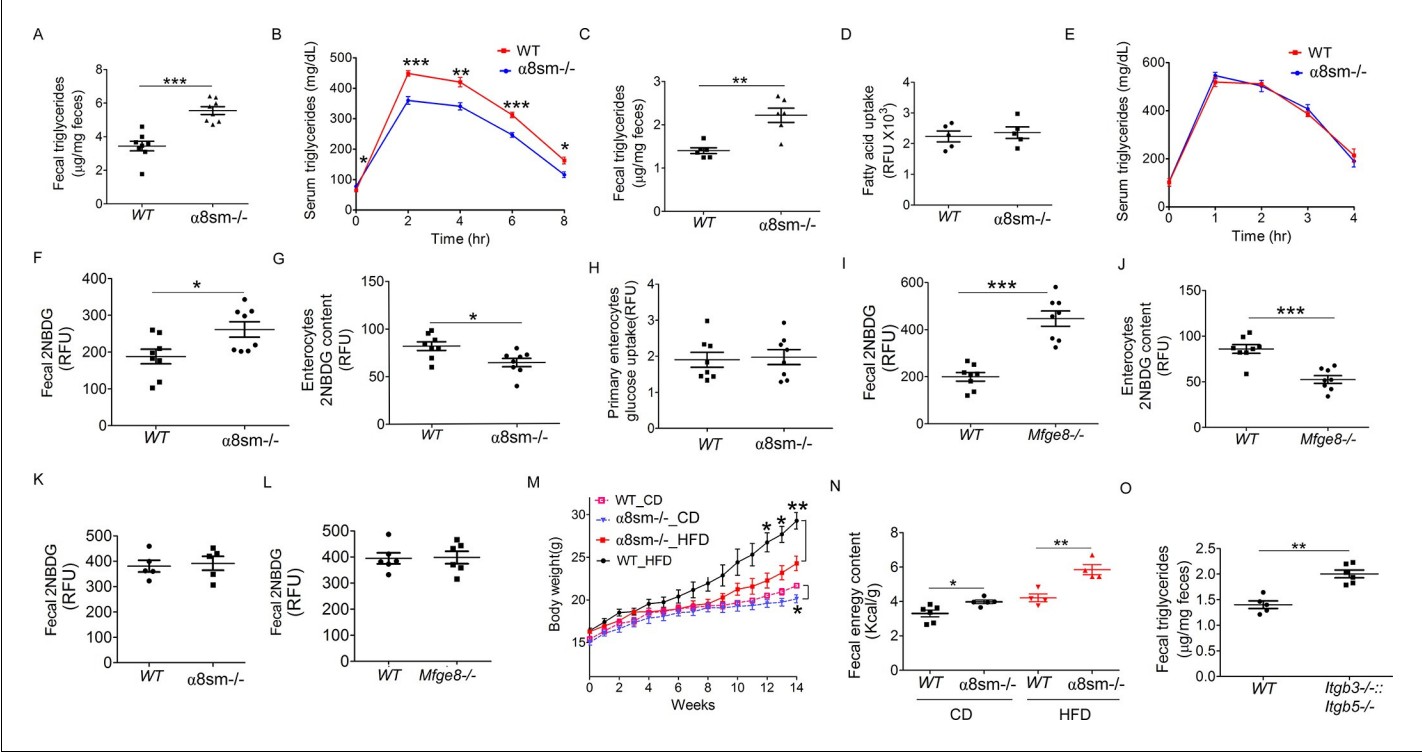

**Figure 6.** α8sm⁻/⁻ mice are protected from diet-induced obesity. (**A**) Fecal triglycerides in WT and α8sm⁻/⁻ mice after an olive oil gavage (N = 8). (**B**) Serum triglycerides levels in WT and α8sm⁻/⁻ mice after an olive oil gavage (N = 5). (**C**) Fecal triglycerides in WT and α8sm⁻/⁻ mice on a normal chow control diet (N = 6). (**D**) Primary enterocyte fatty acid uptake in the isolated enterocytes from WT and α8sm⁻/⁻ mice (N = 5). (**E**) Serum triglycerides levels after IP administration of olive oil in WT and α8sm⁻/⁻ mice (N = 5). (**F, G**) Fecal (F, N = 8) and enterocytes (G, N = 8) 2NBDG content in WT and α8sm⁻/⁻ mice after gavage with a 2NBDG -methylcellulose mixture. (**H**) Glucose uptake assay in isolated primary enterocytes from WT and α8sm⁻/⁻ mice (N = 8). (**I,J**) Fecal (I, N = 8) and enterocytes (J, N = 8) 2NBDG content in WT and *Mfge8⁻/⁻* mice after gavage with a 2NBDG-methylcellulose mixture. (**K, L**) Fecal 2NBDG content in WT and α8sm⁻/⁻(K, N = 5) and *Mfge8⁻/⁻*(L, N = 6) after gavage with a 2NBDG in PBS. (**M**) Weight gain in female WT and α8sm⁻/⁻ mice on a normal chow diet (CD) (N = 6–8) or HFD (N = 8–12). (**N**) Fecal energy content in WT and α8sm⁻/⁻ mice on a normal chow diet (CD) (N = 5–6) or HFD (N = 4–5). Each sample represents stool combined from 3 mice. Female mice were used for all experiments. (**O**) Fecal triglycerides in WT and *Itgb3⁻/⁻::Itgb 5⁻/⁻* integrin-deficient mice with normal chow control diet (N = 5–6). For all in vivo experiments, each group of 5 mice represents 1 independent experiment. *p<0.05, **p<0.01, ***p<0.001. Data are expressed as mean ± s.e.m.

The following figure supplement is available for figure 6:

**Figure supplement 1.** Protection from weight gain in α8sm⁻/⁻ mice on a HFD.

2NBDG levels (*Figure 6I and J*) when 2NBDG was gavaged as a semisolid mixed with methylcellulose but not when administered as a liquid preparation in PBS (*Figure 6K and L*).

*Mfge8⁻/⁻* mice gain approximately 50% less weight on a high-fat diet (HFD) as compared with WT controls (*Khalifeh-Soltani et al., 2014*). To evaluate the relative contribution of altered motility to this phenotype, we placed α8sm⁻/⁻ mice on a HFD. Both female and male α8sm⁻/⁻ mice were significantly protected from weight gain on a HFD (*Figure 6M*, *Figure 6—figure supplement 1A*). Reduced weight gain on a HFD in α8sm⁻/⁻ mice was associated with reduced body fat as measured by Dexa scanning (*Figure 6—figure supplement 1B*). A modest reduction in body weight was also apparent in α8sm⁻/⁻ mice on a CD as compared with WT controls, became statistically significant at 22 weeks of age (*Figure 6M* and S8A), and was associated with decreased body fat on DEXA scan (*Figure 6—figure supplement 1C*). α8sm⁻/⁻ mice also had increased stool energy content as measured by bomb calorimetry on both a HFD and CD (*Figure 6N*). To further explore the contribution of impaired enterocyte fatty acid uptake to steatorrhea in *Mfge8⁻/⁻* mice, we measured stool TG content of *Itgb3⁻/⁻::Itgb5⁻/⁻* mice. *Itgb3⁻/⁻::Itgb5⁻/⁻* mice on a CD had significantly greater stool TG than WT controls (*Figure 6O*) suggesting that both impaired fatty acid uptake mediated through these

integrins and altered motility mediated by the α8β1 integrin contribute to the development of steatorrhea in in *Mfge8*[-/-] mice.

## Discussion

In this work we identify a key role for the α8β1integrin in promoting nutrient absorption through regulation of gastrointestinal motility. Smooth muscle-specific deletion of α8 results in an increase in gastric antral contractile force, more rapid gastric emptying, and faster small intestinal transit times coupled with impaired absorption of both fats and carbohydrates and increased stool energy content. α8sm[-/-] mice are partially protected from weight gain on a HFD and have reduced body weight on a CD. We further show that the milk protein Mfge8 is a novel ligand for α8β1and that binding of Mfge8 to α8β1 is responsible for the effects of this integrin on gastrointestinal motility.

The α8 integrin forms heterodimers with the β1 integrin and was initially identified in axon tracts where the integrin promotes axonal growth during embryogenesis (*Bossy et al., 1991*). Previous work has shown expression of α8 in both vascular and visceral smooth muscle including the muscularis mucosa of the GI tract (*Schnapp et al., 1995*). In vitro studies suggest that α8 promotes smooth muscle proliferation (*Zargham et al., 2007*) and maintains vascular smooth muscle in a differentiated, contractile, non-migratory phenotype (*Zargham et al., 2007*; *Zargham and Thilbault, 2006*; *Zargham et al., 2007*). In vascular smooth muscle, α8 has been shown to co-immunoprecipitate with RhoA and siRNA knockdown of α8 leads to less membrane localization and thereby less active RhoA (*Zargham et al., 2007*). shRNA-mediated knockdown of α8 in intestinal epithelial cells has also been reported to lead to reduction in active RhoA (*Benoit et al., 2009*). In contrast to these findings (*Zargham et al., 2007*; *Benoit et al., 2009*), we found that in gastric smooth muscle, Mfge8 ligation of α8β1 prevents RhoA activation and that this effect is mediated by opposing the activity of PI3K. It is unclear to us why α8β1 has opposing effects on RhoA activation in different cell types. RhoA becomes activated by GTP loading and reverts to an inactive state when GDP-bound. The activation status of RhoA is regulated by a large family of RhoA GTPase Activating Factors (RhoGAPs) and RhoA GTPase guanine nucleotide exchange factors (RhoGEFs) which convert GTP to GDP or load or RhoA would GTP respectively (*Puetz et al., 2009*). We speculate that the most likely explanation for α8β1 preventing RhoA activation in gastric smooth muscle while presumably favoring RhoA activation in vascular smooth muscle and intestinal epithelial cells is differential effects of the α8β1 on and/or differential expression of RhoGAPs and RhoGEFs in specific cell types. These data suggest that the role for the α8β1 integrin in regulation of RhoA and contractility can vary depending on the context within which the integrin is activated. Support for an α8β1-PI3K-RhoA axis regulating motility is evidenced by the ability of the PI3K inhibitor wortmannin to abrogate the exaggerated antral contraction in *Mfge8*[-/-] and α8sm[-/-] antral rings coupled with the increase in AKT phosphorylation and RhoA activation in *Mfge8*[-/-] and α8sm[-/-] antral rings. We also found that PI3K inhibition after Mfge8 ligation of α8β1 occurs through an increase in the activity of the phosphatase PTEN which directly opposes PI3K by converting PIP3 to PIP2. Furthermore, PTEN activity in antral samples had an inverse correlation with the extent of gastric emptying and rate of SIT suggesting an essential role for PTEN in regulating gastrointestinal motility.

Identification of α8β1 as a new binding partner for Mfge8 that slows gastrointestinal motility coupled with malabsorption in α8sm[-/-] mice indicate that the effect of Mfge8 on promoting absorption of dietary fats occurs through dual cooperative mechanisms. Mfge8 slows gastrointestinal motility thereby increasing the time for absorption of dietary fats and directly induces enterocyte fatty acid uptake (*Khalifeh-Soltani et al., 2014*). Since Mfge8 is a secreted molecule, we cannot rule out an effect on antral contraction secondary to Mfge8 binding of integrin receptors on local neurons in the gastrointestinal tract. However, the fact that α8sm[-/-] mice phenocopy the motility phenotype of *Mfge8*[-/-] mice strongly suggests that the dominant effect of Mfge8 on antral contractility is mediated through ligation of integrin receptors on smooth muscle. The relative contribution of altered motility versus impaired fatty acid uptake to the malabsorption phenotype in *Mfge8*[-/-] mice is difficult to parse out from our data. After 12 weeks on a HFD, *Mfge8*[-/-] gained approximately 50% less weight than WT controls (*Khalifeh-Soltani et al., 2014*) while α8sm[-/-] mice gained approximately 40% less weight than WT controls. One interpretation of this data is that the dominant mechanism underlying protection from weight gain in *Mfge8*[-/-] mice on a HFD is accelerated motility. However the elevated stool TG content in *Itgb3*[-/-]::*Itgb5*[-/-] mice suggests a critical contribution of lipid absorption to

steatorrhea since these mice have impaired enterocyte fatty acid uptake but normal motility. Another possibility is that ligands in addition to Mfge8 can bind $\alpha_8\beta_1$ leading to inhibition of gastrointestinal motility and resulting in a relatively greater malabsorption phenotype secondary to altered motility in α8sm[-/-] mice than in Mfge8[-/-] mice. Of note, a recent publication looking at global α8 null mice and global α8 null mice in the ApoE background assessed body weights in each of these mouse lines (**Menendez-Castro et al., 2015**). Consistent with our findings in α8sm[-/-] mice on a control diet, global α8 null mice and global α8 null mice in the ApoE background had reduced body weight as compared with control mice. In global α8 null mice, there was no statistically significant difference in body weights (by our calculation a P value of 0.059), but it is important to note that these mice do not appear to have reached the age at which we saw significant differences in body weights in α8sm[-/-] mice (**Menendez-Castro et al., 2015**). Furthermore, though no statistical significance was reported in the comparison of body weights of the global α8 null mice in the ApoE background at 1 year of age, a 2-tailed t-test of the presented data comparing mice homozygous for the α8 null mutation with wild type mice (both in the ApoE no background) produced a P value of 0.016 (**Menendez-Castro et al., 2015**).

Consistent with accelerated motility as the cause of malabsorption, α8sm[-/-] mice had increased stool triglyceride and 2NBDG levels after gavage with olive oil/2NBDG respectively. Unlike primary enterocytes from Mfge8[-/-] mice, primary enterocytes from α8sm[-/-] mice did not have impaired fatty acid, further supporting altered motility as the cause of steatorrhea in α8sm[-/-] mice. We previously reported that serum glucose levels after gavage of a liquid glucose/PBS mixture were similar in Mfge8[-/-] and WT mice (**Khalifeh-Soltani et al., 2014**). To assess whether exaggerated antral contraction in Mfge8[-/-] mice led to impaired absorption of glucose, we administered 2NBDG in a mixture with methylcellulose. Unlike the previously used liquid glucose mixture (**Khalifeh-Soltani et al., 2014**), the 2NBDG/methylcellulose provides a semisolid substrate that depends on antral contraction for propulsion along the small intestinal tract. As with α8sm[-/-] mice, Mfge8[-/-] mice had increased stool 2NBDG levels. However, when we gavaged mice with 2NBDG in a liquid form in PBS, stool 2NBDG levels were similar in Mfge8[-/-] mice as compared with WT control mice suggesting that the effect of Mfge8 on enterocyte glucose uptake is through altered motility rather than direct uptake.

One interesting observation from our work is the opposing effects Mfge8 has on PI3K activation in smooth muscle cells as compared with adipocytes. Mfge8 slows gastrointestinal motility by increasing PTEN activity leading to inhibition of PI3K activity while Mfge8 increases fatty acid uptake by activating PI3K (**Khalifeh-Soltani et al., 2014**). The effect on fatty acid uptake and PI3K activation is mediated through the αvβ3 and αvβ5 integrins (**Khalifeh-Soltani et al., 2014**) while the effect on gastrointestinal motility and PI3K inhibition is mediated through the α8β1integrin. Of note, both the αvβ3 and αvβ5 integrins are expressed on gastric smooth muscle. However, even though Mfge8 is a ligand for both of these integrins, single and double knockouts of αvβ3 and αvβ5 did not have the gastric smooth muscle phenotype of Mfge8[-/-] mice, suggesting that even if Mfge8 binds these integrins on smooth muscle cells, binding does not lead to alterations in calcium sensitivity and smooth muscle contractility. Furthermore, in relation to modulation of PI3K activity, our data suggest that in smooth muscle cells the effect of Mfge8 ligation of α8β1is dominant over any effect that Mfge8 ligation of αvβ3 and αvβ5 may havesince baseline phosphorylated AKT levels (reflecting PI3K activation) are increased in antral tissue from Mfge8[-/-] mice as compared with WT controls. Furthermore, rMfge8 treatment reduces AKT phosphorylation in antral rings while it increases AKT phosphorylation in 3T3-L1 adipocytes (**Khalifeh-Soltani et al., 2014**). Finally, we have ruled out the possibility that Mfge8 is a ligand for any other RGD-binding integrin (**Atabai et al., 2009**) other than αvβ1 and non-integrin receptors that have not been identified for Mfge8.

Another interesting observation that comes from our data is that oral administration of Mfge8 by gavage successfully prolongs small intestinal transit time and reduces the extent of gastric emptying. We have shown that recombinant Mfge8 reaches the smooth muscle by immunofluorescence. However, we are not entirely sure how Mfge8 gets to the smooth muscle layer. One possibility is that it is absorbed into the bloodstream and takes a hematogenous route to the smooth muscle. In our previously published work, oral Mfge8 gavage did not lead to discernible serum Mfge8 levels 30 min after gavage (**Khalifeh-Soltani et al., 2014**). While this would argue against a hematogenous route to the smooth muscle, it is possible that blood levels are below the sensitivity of the Mfge8 ELISA. Alternatively, measurement of serum levels at earlier time points may show recombinant protein in the bloodstream. Another possibility is that the epithelium takes up the recombinant Mfge8 and

secretes it from the basal side and that Mfge8 reaches the smooth muscle either directly or through entering the circulation on the basal side.

From a therapeutic viewpoint, our findings provide new targets for treatment of diseases associated with altered gastrointestinal motility. Activation of the pathway we describe with recombinant Mfge8 could be used to treat disorders characterized by increased gastrointestinal transit time and malabsorption such as short bowel syndrome. Blockade of the α8β1 integrin could be used as a treatment for gastroparesis secondary to conditions such as diabetic neuropathy, medication side effects, and a number of neurological diseases. Dissociation of the dual effects of Mfge8 at the receptor level provides flexibility for designing therapeutic strategies that can target only the effect on fatty acid uptake or gastrointestinal motility or both simultaneously. In addition, our data provides preliminary proof of principle evidence to support evaluation of α8 blockade as a therapy for gastroparesis.

# Materials and methods

## Mice

All animal experiments were approved by the UCSF Institutional Animal Care and Use Committee in adherence to NIH guidelines and policies. All mice were maintained on a C57BL/6J background. $Mfge8^{-/-}$ mice were obtained from RIKEN (**Hanayama et al., 2002**). Tg(TetO-cre)1Jaw/J and Tg (Acta2-rtTA)#Des mouse lines have been described previously (**Perl et al., 2002**; **Chen et al., 2012**). $Mfge8^{-/-}sm^{+}$ transgenic mice were created by cloning the Mfge8 long isoform into the PTRE2 vector with subsequent microinjection of DNA by the Gladstone Institute Gene-Targeting Core. Tg(TetO-Mfge8) transgenic mice containing the tetracycline-inducible Mfge8 construct were crossed with a $Mfge8^{-/-}$ mice line created using a gene disruption vector (**Atabai et al., 2005**; **Silvestre et al., 2005**) and mice carrying the Tg(Acta2-rtTA)#Des transgene. $Itga8^{flox/flox}$ mice have been previously described (**Chan et al., 2010**). $α8sm^{-/-}$ mice were created by crossing $Itga8^{flox/flox}$ mice with mice carrying Tg(TetO-cre)1Jaw/J and Tg(Acta2-rtTA)#Des transgenes. $Itgb3^{-/-}$, $Itgb5^{-/-}$, and $Itgb3^{-/-}::Itgb5^{-/-}$ mice in the 129 SVEV strain have been previously described (**Huang et al., 2000**; **Sugimoto et al., 2012**). For smooth muscle induction of Mfge8 or Cre- mediated recombination of $Itga8^{flox/flox}$—Tg(Acta2-rtTA, TetO-Cre) mice were placed on doxycycline chow for 2 weeks prior to experiments.

## Cell lines

Human gastric smooth muscle cells were obtained from ScienCell Research Laboratories and have been characterized by immunofluorescent method with antibodies to α-smooth muscle actin and desmin. Cells were received at passage 1 and are negative for HIV-1, HBV, HCV, mycoplasma, bacteria, yeast and fungi. Experiments were conducted with cells between passages 3 and 4. SW480 cells were generously provided by Yasuyuki Yokosaki and Dean Sheppard. SW480 cells were characterized by STR profiling and isoenzyme analysis by ATCC, and were negative for mycoplasma.

## Antral ring contraction

We suspended freshly isolated antral ring slices (2–3 mm in length) on plexiglass rods in a double-jacketed organ bath (Radnoti 8 unit tissue organ bath system) in Krebs-Henseleit solution maintained at 5% CO2-95% O2, 37°C, and a pH of 7.4–7.4533. We attached rings by a silk thread to a FT03 isometric transducer. Concentration-response curves of multiple chambers were continuously displayed and recorded. We set initial tension at 0.5 g for antral rings before adding contractile agonists. We then added a range of concentrations of MCh ($10^{-4}$ to $10^{-9}$ M) and KCl (3.75–60 mM) to induce contraction. For selected studies, wortmannin (100 ng/ML), Y-27632 (100 nm) or recombinant Mfge8 constructs (10 µg/ml) were added 15 min prior to addition of contractile agonists. In the α8 blocking antibody experiments (in *Figure 2J*) we injected 10 mg/kg IP α8 blocking antibody or I IgG1 isotype control antibody (Cell signaling, 5415) 30 min before the antral ring contraction assay.

## Gastric emptying measurement

Mice were fasted for 12 hr prior to experiments but had free access to water. Mice were gavaged with 250 µl of methylcellulose mixed with phenol red (0.5 g/L phenol red in 0.9% NaCl with 1.5%

methylcellulose). We euthanized mice 15 min after administration of the test meal, dissected out the stomach and removed the stomach after ligation of the cardiac and pyloric ends to ensure that any retained meal did not leak out of the stomach during removal. We then cut the stomach into pieces and homogenized with 25 ml of 0.1 N NaOH and added 0.5 ml of trichloroacetic acid (20% w/v) and centrifuged at 3000 rpm for 20 min. We then added 4 ml of 0.5 N NaOH to 1 ml of the supernatant and measured absorbance at 560 nm to assess phenol red content in the stomach. The percentage gastric emptying was derived as (1-X/Y)*100 where X represents absorbance of phenol red recovered from the stomach of animals sacrificed 15 min after a test meal. Y represents mean (n = 5) absorbance of phenol red recovered from the stomachs of control animals which were euthanized immediately following gavage with the test meal. In experiments using rMfge8 and RGE constructs, we administered each construct by gavage (50 µg/kg body weight in a total volume of 200 µl in PBS) before administration of phenol to mice. Y-27632 was administered IP (100 nm) 15 min prior to gavage. In the α8 blocking antibody experiments (in *Figure 2K*) we injected 10 mg/kg IP α8 blocking antibody 30 min before administration of phenol red to mice.

## Small intestinal transit (SIT)

Mice were fasted for 12 hr prior to experiments but had free access to water. We then gavaged mice with 250 µl Carmine meal (6% Carmine red and 0.5% methylcellulose in water). 15 min after administration of gavage, we euthanized mice and dissected out the small intestine from the pylorus to the ileocecal junction, identifying the location to which the meal had traversed, and securing that position with thread to avoid changes in the length of the transit due to handling. The small intestinal transit (SIT) was calculated from the distance traveled by Carmine meal divided by total length of the small intestine multiplied by 100. In experiments using rMfge8 and RGE constructs, we administered each construct by gavage (50 µg/kg body weight in a total volume of 200 µl in PBS) before administration of the Carmine meal to mice. Y-27632 was administered IP (100 nm) 15 min prior to gavage. In the α8 blocking antibody experiment (in *Figure 2L*) we injected 10 mg/kg IP α8 blocking antibody 30 min before administration of Carmine meal to mice.

## Primary enterocytes isolation

We collected primary enterocytes by harvesting the proximal small intestine from anesthetized mice, emptying the luminal contents, washing with 115 mM NaCl, 5.4 mM KCl, 0.96 mM NaH2PO4, 26.19 mM NaHCO3 and 5.5 mM glucose buffer at pH 7.4 and gassing for 30 min with 95% O2 and 5% CO2. We then filled the proximal small intestine with buffer containing 67.5 mM NaCl, 1.5 mM KCl, 0.96 mM NaH2PO4, 26.19 mM NaHCO3, 27 mM sodium citrate and 5.5 mM glucose at pH 7.4, saturated with 95% O2 and 5% CO2, and incubated in a bath containing oxygenated saline at 37°C with constant shaking. After 15 min, we discarded the luminal solutions and filled the intestines with buffer containing 115 mM NaCl, 5.4 mM KCl, 0.96 mM NaH2PO4, 26.19 mM NaHCO3, 1.5 mM EDTA, 0.5 mM dithiothreitol and 5.5 mM glucose at pH 7.4, saturated with 95% O2 and 5% CO2, and we placed them in saline as described above. After 15 min, we collected and centrifuged the luminal contents (1,500 r.p.m., 5 min, room temperature) and resuspended the pellets in DMEM saturated with 95% O2 and 5% CO2).

## Olive oil/2NDGB gavage

We fasted 6- to 8-week-old mice for 4 hr and then each mouse received an oral gavage of 200 µl olive oil or 2 µg per g body weight 2NBDG and 2 µg per g body weight rhodamine-PEG (Methoxyl PEG Rhodamine B, MW 5,000 g mol$^{-1}$) with 0.2% fatty acid–free BSA by gavage. We collected feces from 20 min to 4 hr after 2NBDG was administered. We homogenized 50 mg of feces in PBS containing 30 mM HEPES, 57.51 mM MgCl2 and 0.57 mg ml$^{-1}$ BSA with 0.5% SDS and sonicated for 30 s; we then centrifuged at 1000 g for 10 min. We transferred supernatants to 96-well plates and measured fluorescence values immediately using a fluorescence microplate reader for endpoint reading (Molecular Devices). We then subtracted baseline fluorescence from untreated mice from measured fluorescence. We also measured enterocytes' 2NBDG content after isolation of primary cells as described above, using excitation and emission wavelengths of 488 nm and 515 nm, respectively. For rhodamine-PEG, the excitation and emission wavelengths were 575 nm and 595 nm, respectively.

## Serum and fecal triglycerides measurement

We fasted 6–8 week old mice for 4 hr and administered 200 µl olive oil by oral gavage or IP injection and collected tail vein blood at indicated times. Serum TG concentration was determined by Wako L-Type TG determination kit (Wako Chemicals USA). We collected the feces from 20 min to 4 hr after Olive oil administration. 50 mg of feces were homogenized with chloroform/methanol (2:1) in a 20:1 v/w ratio, the whole mixture was incubated overnight at 4°C with gentle shaking. Then, 0.2 volume of 0.9% NaCl was added and centrifuged at 500 g for 30 min After extracting the organic phase, samples were evaporated under nitrogen until dry and reconstituted in PBS containing 1% Triton X-100 for TG measurement by Wako L-Type TG determination kit (Wako Chemicals USA).

## Solid phase binding assay

Direct binding of Mfge8 with α8 was assessed by solid-phase binding in non-tissue coated microplates. Either recombinant α8, αvβ3, or α5β1 were attached to the plates and purified Mfge8 was added for 2 hr at room temperature in the presence or absence of 10mM EDTA. For α5β1, 1 mM $MgCl^{2+}$ and 1 mg/mL $CaCl^{2+}$ was added to activate β1. Following 5 washes with PBS + 1% BSA and 0.05% Tween, the extent of Mfge8 binding was detected using a biotinylated antibody against Mfge8 (1:1000, 1 hr at 37°C). Then streptavidin-HRP was added for 20 min at room temperature followed by 3,3′,5,5′ tetramethylbenzidine substrate solution. Absorbance was then measured at 450 nm in a microplate reader.

## Cell adhesion assay

Cell adhesion assays were performed as described (*Yokosaki et al., 1994*) with slight modifications. Briefly, $1 \times 10^5$ cells were seeded into each well of 96 well MaxiSorp enzyme-linked immunosorbent assay plates (Nunc) coated with substrate proteins at 37°C for 1 hr and then incubated for 1 hr at 37°C. Attached cells were stained with 0.5% crystal violet and solubilized in 2% Triton X-100 for taking optical density at 595 nm. For blocking experiments, cells were incubated with antibodies (5 µg/ml) before plating for 15 min on ice.

## Human gastric smooth muscle cells siRNA treatment

HGSMCs were obtained from commercial sources (Science Cell Research Laboratories) and maintained in minimum essential medium supplemented with 10% FBS at 37°C with 5% CO2. We plated the cells in six-well plates 1 day prior to infection. We transfected cells with 100 nM PTEN siRNA (ON-TARGETplus Human PTEN, Thermo Fisher Scientific) or controls (ON-TARGETplus Scramble Control siRNA, Human, Thermo Fisher Scientific) in antibiotic- and norepinephrine-free culture medium using Lipofectamine-2000 (Invitrogen). 6 hr later, we change the medium to fully supplemented medium and conducted assays 48 hr after transfection.

## Immunofluorescent microscopy

Mice were starved for 4 hr before gavage with recombinant Mfge8 (50 µg/kg). Antrum samples were removed 30 min post-gavage and fixed with paraformaldehyde and paraffin-embedded or immediately frozen. Sections (5–10 µm) were cut and stained for α-smooth muscle actin (Sigma-Aldrich), integrin α8 (YZ3) and human FC (Rockland). The recombinant Mfge8 consists of a fusion protein containing full length Mfge8 and a human Fc domain (*Atabai et al., 2009*).

## Quantification of muscle thickness

Five images were taken per antrum section and each image was divided into fifths. Thickness of individual muscle layers was quantified at each point using a scale of 314 pixels to 100 µm. Averaged thickness is reported in *Figure 1—figure supplement 1*.

## RhoA activation assay

The RhoA activation assay was performed according to the manufacturer's instructions (Cytoskeleton). Briefly, we dissected out the gastric antrum, gently removed the mucosal layer and incubate them with methacholine (100 µm) for 15 min and then homogenized the muscle layer in lysis buffer (50 mM Tris-HCl, pH 7.5, 10 mM MgCl2, 0.5 M NaCl, 1% Triton X-100, and protease and

phosphatase inhibitor cocktail (Thermo). Human Gastric Smooth Muscle Cells (HGSMC) were treated with 5-hydroxytryptamine (100 µM) for 15 min and then lysed. We collected the supernatants after centrifugation and incubated with GST-Rhotekin bound to glutathione-agarose beads at 4°C for 1 hr. We washed the beads with a wash buffer containing 25 mM Tris, pH 7.5, 30 mM MgCl2, and 40 mM NaCl. GTP-bound RhoA was detected by immunoblotting.

## PTEN activity assay

We isolated antral lysates or human gastric smooth muscle cell lysates and measured conversion of PI(3,4,5) P3 to PI(4,5)P2 (PTEN activity ELISA, Echelon) after incubation with recombinant proteins (rMfge8, RGE, Fibronectin, or Vitronectin, R&D Systems, Inc. 10 µg/ml) or blocking antibodies against α8, β3, and β5 (10 g/ml). In this competitive ELISA, we incubated lysates on a PI(4,5)P2 coated microplate and then added a PI(4,5)P2 detector protein. PI(4,5)P2 produced by PTEN in lysate binds to the detector protein and thus prevents it from binding immobilized PI(4,5)P2 on the plate. We then used a peroxidase-linked secondary to measure PI(4,5)P2 detector protein binding to the plate in a colorimetric assay where the colorimetric signal is inversely proportional to the amount of PI(4,5)P2 produced by PTEN.

## Western blots

We lysed tissues in cold RIPA buffer (50 mM Tris HCl pH 7.5, 150 mM NaCl, 1% NP-40, 1% sodium deoxycholate, 0.1% SDS) supplemented with complete miniprotease and phosphatase inhibitor cocktail (Pierce, Rockford, IL). We incubated lysates at 4°C with gentle rocking for 30 min, sonicated on ice for 30 s (in 5 s bursts) and then centrifuged at 12,800 rpm for 15 min at 4°C. We determined protein concentration by Bradford assay (Bio-Rad, Hercules, CA). We separated 20 µg of protein by SDS-PAGE on 7.5% resolving gels (Bio-Rad) and transblotted onto polyvinylidene fluoride membranes (Millipore). We incubated the membranes with a 1:1,000 dilution of antibodies against Akt (catalog 9272, Cell Signaling), phospho-Akt Ser473 (clone 193H12, Cell Signaling), MLC (catalog 3672S Cell Signaling) phospho-MLC (clone 519, Cell Signaling), MYPT (catalog 2634S, Cell Signaling) phospho-MYPT (catalog 5163, Cell Signaling), RhoA (clone 67139, Cell Signaling), PTEN (clone 138G6, Cell Signaling), or GAPDH (clone 14C10, Cell Signaling) followed by a secondary HRP-conjugated antibody. For evaluation of total Akt, MLC or MYPT we stripped and reprobed membranes that had been blotted for phospho-versions of these proteins. Blots were developed using the enhance chemical luminescence system (Amersham).

## Recombinant protein production

We created and expressed recombinant Mfge8 and RGE protein constructs in High Five cells as previously described (*Atabai et al., 2009*). All constructs were expressed with a human Fc domain for purification across a protein G sepharose column. For experiments in *Figure 3A and B*, Mfge8 was expressed in Freestyle 293 cells with His-tag and purified by Ni-NTA column.

## High-fat diet

We placed 8-week-old α8sm$^{-/-}$ mice on a high-fat diet formula containing 60% fat calories (Research Diets) for 12 weeks. Mouse were placed on doxycycline chow (2 g/kg, Bioserve) for two weeks prior to beginning the HFD and subsequently had doxycycline in their water (0.2 mg/ml) for the duration of the experiments.

## Body composition analysis

We performed bone, lean and fat mass analysis with a GE Lunar PIXImus II Dual Energy X-ray Absorptiometer.

## Measurements of fecal energy content

We freeze-dried feces from mice on a HFD and pulverized them with a ceramic mortar and pestle. We measured caloric content of feces with an 1108 Oxygen Combustion Bomb calorimeter.

## Statistical analyses

We assessed data for normal distribution and similar variance between groups using GraphPad Prism 6.0. We used a one-way ANOVA to make comparisons between multiple groups. When the ANOVA comparison was statistically significant ($P < 0.05$), we performed further pairwise analysis using a Bonferroni t-test. We used a two-sided Student's t-test for comparisons between 2 groups. For analysis of weight gain over time in mice, we used a two-way ANOVA for repeated measures. We used GraphPad Prism 6.0 for all statistical analyses. We presented all data as mean $\pm$ s.e.m. We selected sample size for animal experiments based on numbers typically used in the literature. We did not perform randomization of animals.

## Acknowledgements

This research was supported by US National Institutes of Health grant P30 DK 063-7202 and the University of California, San Francisco (UCSF) Diabetes and Endocrinology Research Center and UCSF Cardiovascular Research Institute startup funds (KA), and an Award from the American Heart Association Western States Affiliate and the American Brain Foundation (#14POST18580033) (AKS). We would like to thank Amha Atakilit and Dean Sheppard (UCSF Lung Biology Center) for providing integrin-blocking antibodies. We would like to thank Stephen Layer for ongoing inspiration.

## Additional information

### Funding

| Funder | Author |
|---|---|
| National Institute of Diabetes and Digestive and Kidney Diseases | Kamran Atabai |
| American Heart Association | Amin Khalifeh-Soltani |
| American Brain Foundation | Amin Khalifeh-Soltani |
| American Stroke Association | Amin Khalifeh-Soltani |

The funders had no role in study design, data collection and interpretation, or the decision to submit the work for publication.

### Author contributions

AK-S, AH, KA, Conception and design, Acquisition of data, Analysis and interpretation of data, Drafting or revising the article; MJP, Analysis and interpretation of data, Drafting or revising the article; DAM, DH, Conception and design, Analysis and interpretation of data, Drafting or revising the article; WM, Conception and design, Acquisition of data, Analysis and interpretation of data; SA, Acquisition of data, Analysis and interpretation of data, Drafting or revising the article; SS, KMT, NW, Acquisition of data, Analysis and interpretation of data, Contributed unpublished essential data or reagents; YY, AS, Conception and design, Acquisition of data, Analysis and interpretation of data, Contributed unpublished essential data or reagents

### Author ORCIDs

Kamran Atabai, http://orcid.org/0000-0001-8170-7881

### Ethics

Animal experimentation: All animal experiments were approved by the UCSF Institutional Animal Care and Use Committee in adherence to NIH guidelines and policies.(#AN109941-01A)

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
