## [Decision Letter]

Thank you for submitting your work entitled "α8β1 integrin regulates nutrient absorption through an Mfge8-PTEN dependent mechanism" for consideration by *eLife*. Your article has been reviewed by 3 peer reviewers, and the evaluation has been overseen by the Reviewing Editor Johanna Ivaska and Ivan Dikic as the Senior Editor.

The reviewers have discussed the reviews with one another and the Reviewing Editor has drafted this decision to help you prepare a revised submission.

Summary:

In general all reviewers found the work interesting and potentially suitable for publication in *eLife* provided that the revisions listed below are successfully completed.

Essential revisions:

1) The authors should provide some direct evidence by performing experiments such as in-vitro binding studies to support their claim for a specific interaction of Mfge8 with the Integrin α8β1 and compare the interaction of Mfge8 with other Integrins. Previous studies have implicated Mfge8 interaction with other combinations of Integrin receptors. While these seem to have been well cited, the possibility of Mfge8 as a ligand for specific or multiple Integrins should be discussed, along with a rationale for why one particular interaction might be relevant for gastric smooth muscles.

2) Please demonstrate whether Mfge8 and α8 integrin co-localize in antral smooth muscles, in support of the notion that they physically interact.

3) Can the authors rule out an indirect effect on antral smooth muscle contraction due to interaction of Mfge8 with Integrins present on local neurons? Such a possibility should be experimentally addressed and/or discussed.

4) Do smooth muscles develop normally in Mfge8^-/-^ mice, given the absence of Mfge8 during neonatal development? Histological or other evidence should be provided to rule out potential alternate mechanisms such as muscle hypertrophy that might also explain enhanced muscle contractility in these mice.

5) The authors should also discuss how Mfge8 that is externally administered via gavaging could penetrate the epithelial and mucosal layers to access smooth muscle Integrin receptors at a deeper location.

6) PTEN activity assay in Figure 5 needs to show conversion of PIP3 to PIP2 using TLC or some other appropriate assay. PIP2 levels can vary in response to activity of many different enzymes, therefore extrapolation of PIP2 levels alone to PTEN activity is not sufficient.

7) Farias et al. (BBRC, 2005) showed that signaling via α8 integrin promoted PI3 kinase activation. In this study the authors describe that signaling via α8 integrin inhibits PI3 kinase activity. This should be discussed.

8) α8sm^-/-^ mice of this study are protected from weight gain compared to wild types even on control diet after 14 weeks. On the other hand the weights of α8 integrin deficient mice or α8 integrin deficient mice crossed with ApoE deficient mice do not differ from wild types as shown by Menendez-Castro et al. (J Pathol 2015) This should be discussed.

---

## [Author Response]

Essential revisions:

1) The authors should provide some direct evidence by performing experiments such as in-vitro binding studies to support their claim for a specific interaction of Mfge8 with the Integrin α8β1 and compare the interaction of Mfge8 with other Integrins.

We have now included solid-phase binding studies to show a direct interaction between Mfge8 and the α8β1 integrin (Figure 2).

Previous studies have implicated Mfge8 interaction with other combinations of Integrin receptors. While these seem to have been well cited, the possibility of Mfge8 as a ligand for specific or multiple Integrins should be discussed, along with a rationale for why one particular interaction might be relevant for gastric smooth muscles.

Mfge8 is known to bind both the αvβ3 and αvβ5 integrins. Therefore our first investigations were focused on understanding whether these integrins mediated the effect of Mfge8 on smooth muscle contraction. As presented in Figure 2—figure supplement 2 and Figure 5—figure supplement 1, we did an extensive in vivo evaluation of both single and double knockouts of each of these integrins and found that the knockouts did not phenocopy the Mfge8 knockout. Of the remaining RGD binding integrins, we have previously published that Mfge8 does not bind αvβ6, αvβ8 or α5β1 (Atabai et al., J Clin Invest 2009) and therefore focused on α8β1 given its extensive smooth muscle expression. While it is possible (and likely) that Mfge8 binds αvβ3 and αvβ5 on smooth muscle, our data supports the conclusion that the effect on smooth muscle calcium sensitivity and contraction is mediated through ligation of 1. It is interesting to note that while we show in this work that Mfge8 ligation of α8β1 leads to inhibition of PI3 kinase activity, we have previously shown that Mfge8 binding of the αvβ3 and αvβ5 increases PI3 kinase activity in enterocytes (Khalifeh-Soltani et al., Nat Med 2014). Therefore one possibility is that the effect of α8β1 in smooth muscle on PI3 kinase activity dominates any effect seen through ligation of αvβ3 and αvβ5. Alternatively, there may be different adapter proteins at play in enterocytes than in smooth muscle cells that dictate how integrin ligation modulates PI3 kinase activity. We have expanded our discussion of this point in the Discussion section of the manuscript (second paragraph).

2) Please demonstrate whether Mfge8 and α8 integrin co-localize in antral smooth muscles, in support of the notion that they physically interact.

We have performed immunofluorescence experiments on antral tissue isolated from Mfge8 KO and α8 smooth muscle KO mice that show that after gavage with recombinant Mfge8, double immunofluorescent staining demonstrates co-localization of Mfge8 with α8 in antral smooth muscle (Figure 2). Please note that we used gavage with recombinant Mfge8 in the Mfge8 KO background because of the relatively poor anti-Mfge8 antibodies available for immunofluorescence. The recombinant Mfge8 allowed us to target the fused human-Fc domain with an anti-Fc antibody which provides more reliable staining.

3) Can the authors rule out an indirect effect on antral smooth muscle contraction due to interaction of Mfge8 with Integrins present on local neurons? Such a possibility should be experimentally addressed and/or discussed.

Since Mfge8 is a secreted molecule, a tissue-specific knockout or tissue specific-transgenic mouse expressing Mfge8 in the knockout background will not effectively rule out the possibility that in addition to ligation of integrin receptors on smooth muscle, interactions with integrin-expressing local neurons may contribute to the contraction phenotype. However, since mice with smooth muscle-specific deletion of the α8 integrin phenocopy the antral contraction phenotype observed in global Mfge8 knockouts, we can be confident that the dominant effect of Mfge8 on gastric smooth muscle contraction is due to ligation of the α8 integrin on smooth muscle. We have added a paragraph of the discussion addressing this issue (Discussion section, third paragraph).

4) Do smooth muscles develop normally in Mfge8^-/-^ mice, given the absence of Mfge8 during neonatal development? Histological or other evidence should be provided to rule out potential alternate mechanisms such as muscle hypertrophy that might also explain enhanced muscle contractility in these mice.

We have addressed this issue by morphometric analysis of gastric smooth muscle in the adult Mfge8 KO and WT mice. As shown in Figure 1—figure supplement 1, there is no significant increase in smooth muscle thickness, indicating that smooth muscle hypertrophy is not contributing to enhanced contraction in the absence of Mfge8. This is also consistent with our previous evaluation of airway smooth muscle (Kudo et al., Proc Natl Acad Sci USA, 2013).

5) The authors should also discuss how Mfge8 that is externally administered via gavaging could penetrate the epithelial and mucosal layers to access smooth muscle Integrin receptors at a deeper location.

To address this experimentally, we gavaged Mfge8 KO mice with recombinant Mfge8 and then stained for rMfge8 in antral smooth muscle (Figure 2). We found positive staining of exogenously administered recombinant Mfge8 throughout the antral smooth muscle. We speculate that Mfge8 is reaching the smooth muscle either through the bloodstream or by being taken up by the epithelial layer and subsequently being secreted from the basal surface towards the smooth muscle or entering the bloodstream from the basal side. In our previous publication using the same dose of oral Mfge8 (as used in this manuscript) to increase fat absorption, we were unable to measure detectable serum Mfge8 levels by ELISA 30 minutes after gavage^2^. These data would suggest that either oral Mfge8 does not get into the bloodstream or, if it does, it is below the detection level of the currently available ELISA. We have added a paragraph of the discussion addressing this issue (Discussion section, sixth paragraph).

6) PTEN activity assay in Figure 5 needs to show conversion of PIP3 to PIP2 using TLC or some other appropriate assay. PIP2 levels can vary in response to activity of many different enzymes, therefore extrapolation of PIP2 levels alone to PTEN activity is not sufficient.

The PTEN activity ELISA we utilized (Echelon cat # K-4700) specifically detects the amount of the phosphoinositide product (PI(3,4,5)P3 to PI(4,5)P2) produced in tissue lysates. The standard TLC assay also measures conversion of PIP3 to PIP2 and therefore provides no additional information beyond the assay that we used. Furthermore, many of the older assays measure the release of phosphate and therefore are less specific than the assay which we used. We agree that in theory the PIP2 produced in the lysate could be the result of another phosphatase in the tissue. However, PTEN is the main known phosphatase that mediates this reaction. In addition, the fact that siRNA knockdown of PTEN lead to enhanced calcium sensitivity and RhoA activation that was not rescued by the addition of recombinant Mfge8 (Figure 5) gives us further confidence that PTEN is the key phosphatase modulating this pathway.

7) Farias et al. (BBRC, 2005) showed that signaling via α8 integrin promoted PI3 kinase activation. In this study the authors describe that signaling via α8 integrin inhibits PI3 kinase activity. This should be discussed.

We appreciate that our data is in opposition to what has been published in vascular smooth muscle and epithelium. We speculate that this is likely due to differences in intracellular modulators of PI3K activity. As requested, we have added further discussion of this topic in the Discussion section of the manuscript (fifth paragraph).

8) α8sm^-/-^ mice of this study are protected from weight gain compared to wild types even on control diet after 14 weeks. On the other hand the weights of α8 integrin deficient mice or α8 integrin deficient mice crossed with ApoE deficient mice do not differ from wild types as shown by Menendez-Castro et al. (J Pathol 2015) This should be discussed.

We have addressed this issue in the Discussion section (third paragraph). In the paper by Menendez-Castro (Menendez-Castro et al., J Pathol 2013), the authors evaluated global α8 knockout mice as well as α8 knockout mice in the ApoE null background. As best as we can tell from the Methods section, the weights reported in table 1 for the global α8 knockouts are from mice that are likely between 10-16 weeks of age. We are making this estimation based on the fact that the authors state that male mice of 18 g in weight were studied and we presume this is in the c57bl/6 background (though this is not specifically stated). Our mice of that strain and sex and weight are typically 6-8 weeks of age. Then, the mice were sacrificed either 4 or 8 weeks after carotid artery ligation, but it is not clear to us which of these time points are presented in table 1. It does seem safe to conclude that the mice at their oldest would have been 16 weeks of age. Though there are no statistically significant differences in weights reported in table 1, there does appear to be a trend towards reduced weight in the α8 global knockout with the homozygotes weighing less than the heterozygotes which weigh less than the wild types. Furthermore, when we perform a 2-tailed t-test on the presented data in table 1 comparing the wild type with the global α8 knockout, we get a P value of 0.059. In our dataset, we only observe a statistically significant difference in body weight in male mice that are 21 weeks of age on a normal chow diet and the difference is quite modest. Therefore, we believe that our data is consistent with the published data. In table 2 of the paper by Menendez-Castro, the authors show that the weights of α8 KOs in the ApoE null background are no different than wild type. However,, there does appear to be a difference in weights with the ApoE null in the wild type background weighing 34.34 g with a standard deviation of 0.8 and the global α8 KOs in the ApoE null background weighing 31.54 g with a standard deviation of 0.65. Of note, these mice were studied at one year of age and the latter group developed evidence for renal disease. Whether renal disease led to retention of fluid thereby influencing body weights was not experimentally addressed and body composition was not reported. Therefore, it is somewhat difficult to compare the results of these body weights with the smooth muscle specific knockouts, we studied. That being said, we believe that the trend towards reduced body weight in the α8 KO in the ApoE null background is consistent with our findings and appeared to be quite close to reaching statistical significance based on the reported standard deviation. In fact, when we perform a 2-tailed t-test comparing ApoE null in the wild type background with the α8 KO in the ApoE null background using the reported body weights, sample size, and standard error of the mean we get a P value of 0.016 indicating that in fact there is statistical significance between body weights.

References:

1) Atabai, K., et al. Mfge8 diminishes the severity of tissue fibrosis in mice by binding and targeting collagen for uptake by macrophages. The Journal of clinical investigation 119, 3713-3722 (2009).

2) Khalifeh-Soltani, A., et al. Mfge8 promotes obesity by mediating the uptake of dietary fats and serum fatty acids. Nature medicine 20, 175-183 (2014).

3) Kudo, M., et al. Mfge8 suppresses airway hyperresponsiveness in asthma by regulating smooth muscle contraction. Proceedings of the National Academy of Sciences of the United States of America 110, 660-665 (2013).

4) Menendez-Castro, C., et al. Under-expression of alpha8 integrin aggravates experimental atherosclerosis. The Journal of pathology 236, 5-16 (2015).